# Enhanced Intestinal TGF-β/SMAD-Dependent Signaling in Simian Immunodeficiency Virus Infected Rhesus Macaques

**DOI:** 10.3390/cells10040806

**Published:** 2021-04-04

**Authors:** Nongthombam Boby, Alyssa Ransom, Barcley T. Pace, Kelsey M. Williams, Christopher Mabee, Arpita Das, Sudesh K. Srivastav, Edith Porter, Bapi Pahar

**Affiliations:** 1Division of Comparative Pathology, Tulane National Primate Research Center, Covington, LA 70433, USA; nboby@tulane.edu (N.B.); aranso@dcc.edu (A.R.); barcleypace@gmail.com (B.T.P.); kwilli30@tulane.edu (K.M.W.); cmabee@tulane.edu (C.M.); 2Division of Microbiology, Tulane National Primate Research Center, Covington, LA 70433, USA; arpita@tulane.edu; 3Department of Biostatistics, Tulane University, New Orleans, LA 70112, USA; ssrivas@tulane.edu; 4Department of Biological Sciences, California State University Los Angeles, Los Angeles, CA 90032, USA; eporter@calstatela.edu; 5Tulane University School of Medicine, Tulane University, New Orleans, LA 70112, USA; 6Tulane School of Public Health and Tropical Medicine, New Orleans, LA 70112, USA

**Keywords:** CD4, CD68, rhesus macaque, simian immunodeficiency virus, TGF-β1, TGF-β receptor, SMAD signaling pathway, SMAD3, SMAD7

## Abstract

Transforming growth factor-β signaling (TGF-β) maintains a balanced physiological function including cell growth, differentiation, and proliferation and regulation of immune system by modulating either SMAD2/3 and SMAD7 (SMAD-dependent) or SMAD-independent signaling pathways under normal conditions. Increased production of TGF-β promotes immunosuppression in Human Immunodeficiency Virus (HIV)/Simian Immunodeficiency Virus (SIV) infection. However, the cellular source and downstream events of increased TGF-β production that attributes to its pathological manifestations remain unknown. Here, we have shown increased production of TGF-β in a majority of intestinal CD3^−^CD20^−^CD68^+^ cells from acute and chronically SIV infected rhesus macaques, which negatively correlated with the frequency of jejunum CD4^+^ T cells. No significant changes in intestinal TGF-β receptor II expression were observed but increased production of the pSMAD2/3 protein and SMAD3 gene expression in jejunum tissues that were accompanied by a downregulation of SMAD7 protein and gene expression. Enhanced TGF-β production by intestinal CD3^−^CD20^−^CD68^+^ cells and increased TGF-β/SMAD-dependent signaling might be due to a disruption of a negative feedback loop mediated by SMAD7. This suggests that SIV infection impacts the SMAD-dependent signaling pathway of TGF-β and provides a potential framework for further study to understand the role of viral factor(s) in modulating TGF-β production and downregulating SMAD7 expression in SIV. Regulation of mucosal TGF-β expression by therapeutic TGF-β blockers may help to create effective antiviral mucosal immune responses.

## 1. Introduction

Transforming growth factor-β1 (TGF-β1) is an immunoregulatory cytokine that is produced by a variety of immune cells including lymphocytes, macrophages, dendritic cells, and intestinal epithelial cells. It has pleiotropic effects on cell migration, differentiation, proliferation, and survival, and is also involved in regulating immune responses [1,2]. Highlighting its role in maintaining intestinal homeostasis and epithelial cell integrity, we have recently demonstrated that TGF-β promotes epithelial cell survival in rhesus macaque (RhM) colon explants via increased pAKT and decreased IFNγ expression [1]. Since it has both adverse and favorable physiological activities on diverse cellular processes, stringent regulation of its signaling pathways is necessary to ensure its proper and balanced physiological functions [3,4]. The active form of TGF-β1 exerts its biological effects by either SMAD-independent or SMAD-dependent pathway [5,6]. SMADs are a group of intracellular proteins that transmit signals from the cell cytoplasm to the nucleus, which may activate or suppress a cascade of gene expressions [7]. The SMAD-dependent signaling pathway is initiated with oligomerization and activation of TGF-β receptor-II (TGF-βRII) upon binding with TGF-β ligand, which further activates TGF-β receptor-I (TGF-βRI) [8]. The activation of TGF-βRI induces phosphorylation and activation of SMAD2 and SMAD3. The SMAD2-SMAD3 complex then binds with SMAD4 and translocates to the nucleus to act as a transcription factor, regulating the expression of different target genes including IL-1β, TNF-α, MCP-1 genes (inflammation), Chitinase-3-like protein 1 genes (fibrosis), Bcl-2, Caspase-3 genes (apoptosis) and Granzyme A, Granzyme B, IFNγ, and FasL genes (cytotoxic activity of CD8^+^ cells) [9,10]. The SMAD2-3/SMAD4 complex binds to SMAD7 promoter site and regulates its expression [11]. SMAD7 negatively regulates the pathway by various mechanisms, either regulating the stability of TGF-βRI [12], inhibiting activity by directly binding with receptor-regulated SMADs leading to their polyubiquitination and proteosomal degradation [13], or interfering in the formation of the SMAD-DNA complex [14]. The SMAD-independent pathway is mediated via various protein kinases of mitogen-activate protein kinase (MAPK) family members like extracellular signal-regulated kinase (ERK), c-Jun N-terminal kinase (JNK), phosphatidylinositol-3-kinase/protein kinase B (PI3K/AKT), and Rho-like GTPase. These are activated by ligand-occupied receptors to repress, modulate, or activate several downstream cellular responses including TGF-β-mediated apoptosis, extracellular matrix production, and differentiation [15].

TGF-β released by lymphoid and non-lymphoid cells promotes immunosuppression in Human Immunodeficiency Virus (HIV)/Simian Immunodeficiency Virus infection through its inhibitory effects on the adaptive immune system and macrophage deactivation [16,17]. TGF-β induces apoptosis of CD4^+^ T lymphocytes by increasing the expression of apoptosis-inducing factor and caspase-3, among others [18]. Furthermore, TGF-β has profibrotic activity and may contribute to a depletion of naïve T cells from secondary lymphoid organs in HIV/SIV infection [19]. This might compromise the repopulation of T cells in the secondary lymphoid organs even after antiretroviral therapy, resulting in chronic immune suppression. In addition to its role in immune system regulation in HIV/SIV infection, several studies have shown a role for TGF-β in different HIV-related comorbidity events like cardiovascular disorder, nephropathy, liver abnormalities, and lung disease [20].

Enhanced TGF-β effects may result from enhanced TGF-β receptor expression or enhanced intracellular signaling. The mechanism mediating increased TGF-β effects in HIV/SIV infection that attributes to its pathological manifestations remains unknown. Since SMAD protein regulates the fate of SMAD-dependent TGF-β-induced biological activities, and since several studies have described SMAD signaling deregulation in association with various pathological conditions like tumor progression [21], neurological disorders [22], tissue fibrosis [19], aberrant immune response [23], and inflammation [24], we hypothesized that interference with the SMAD signaling pathway might also play an important part in TGF-β-induced HIV/SIV pathogenesis. The RhM SIV infection model is a well-accepted model to study HIV pathogenesis [25]. Here, we present evidence that in a RhM SIV infection model, increased production of intestinal TGF-β was not accompanied by increased TGF-β receptor expression; rather, we found increased phosphorylated SMAD2/3 (pSMAD2/3) regulatory protein production and SMAD3 gene expression, along with significantly decreased protein and gene expression of the inhibitory SMAD7 during SIV infection. 

## 2. Materials and Methods

### 2.1. Animals, Ethics Statement, and Tissue Sampling

In this study, 34 adult Indian RhMs (*Macaca mulatta*) were used and screened negative for SIV, HIV-2, type-D retrovirus, and simian T-cell leukemia virus 1 infection at the initiation of the study (Table 1). The animals were maintained under the care and supervision of Tulane National Primate Research Center (TNPRC) veterinarians within the guidelines of the United States Public Health Service Policy and the Guide for the Care and Use of Laboratory Animals. All animal procedures including virus administration, sample collection, and euthanasia were performed under the direction of TNPRC veterinarians and approved Tulane Institutional Animal Care and Use Committee (IACUC) protocol. All animals were housed in Animal Biosafety Level 2 indoor housing with a 50:50 ratio of light/dark cycle, 30–70% humidity, and a room temperature between 64 and 72 °F. Ad libitum water and food was provided, as is standard procedure for all laboratory animals. A total of 34 RhMs of both sexes between 2.5 and 14.1 years of age were grouped into uninfected controls (*n* = 13), acute SIV infection (8–21 days after SIV infection, *n* = 11), and chronic SIV infection (150–422 days after SIV infection, *n* = 10) based on the days after SIV_MAC_251 infection (Table 1). Animals were infected with 100–1000 TCID_50_ SIV_MAC_251 using either intravenous (IV) or intravaginal (IVAG) routes (Table 1). 

In our earlier studies we have not detected any association between viral dosage, CD4 depletion, and viral loads in RhMs [26,27,28]. Based on the stage of menstrual cycle in female RhMs, the vaginal mucosal may exhibit variation in mucosal thickness and a high single dose is preferred to infect female RhMs [28,29]. We also used IV and IVAG routes of infection to mimic the major routes of transmission in HIV infected patients. Serially diluted SIV was grown in CEMx174 cell line and on 14 d after infection, cells were harvested. Cell culture supernatant was run for SIVp27 antigen ELISA and TCID_50_ was determined using the Reed and Muench method. Chronically SIV infected animals were sampled for peripheral blood and jejunum biopsies at certain intervals. At necropsy, peripheral blood and jejunum tissues were collected. Samples were processed for flow cytometry, immunohistochemistry, immunofluorescence, and real time PCR analyses. 

### 2.2. Plasma Viral Load Quantification

Plasma viral RNA was quantified by either a bDNA signal amplification assay (Siemens Diagnostics, Malvern, PA, USA) or quantitative reverse transcription-PCR (RT-PCR) at the Wisconsin National Primate Research Center with a lower detection limit of 125 and 60 SIV RNA copies/mL of plasma, respectively [28].

### 2.3. Isolation and Enrichment of Lymphocytes from Intestine

Lamina propria lymphocytes (LPLs) from the jejunum were isolated as reported by us earlier [28,30]. In brief, jejunum biopsies were obtained by endoscopy, washed properly in chilled sterile PBS, minced with scissors, and digested with EDTA solution in a 37 °C shaker for 30 min to detach the majority of intestinal epithelial cells. After the removal of epithelial cells by filtration through screen cup strainer with 50 mesh size (0.229 mm) (Sigma-Aldrich, St. Louis, MO, USA), the remaining tissues on the strainer were scrapped and further minced with scissors and digested with media containing 60 U/mL of collagenase (type II, Sigma-Aldrich). The cell suspension was passed through a 16-gauge feeding needle for better separation of any clumps. The larger clumps were removed using a nylon biopsy bag (ThermoFisher, Waltham, MA, USA). To enrich LPLs, cells were centrifuged over discontinuous percoll (Sigma-Aldrich) gradients. The interface layer containing LPLs between the two percoll layers was collected, washed in PBS, and resuspended with complete media. This percoll separation removed dead cells and enriched the cell population for lymphocytes. However, these enriched LPLs also contain trace amount of epithelial and other leukocyte positive cells as reported by us elsewhere [31]. The enriched LPLs were used for flow cytometry assays.

### 2.4. Immunohistochemistry Staining

To quantify TGF-β^+^ cells in jejunal tissues, formalin-fixed tissue sections were stained with anti-TGF-β antibodies using the Mach3 Mouse AP-Polymer detection kit (Biocare Medical, Pacheco, CA, USA) as done previously [1]. The paraffin-embedded tissue sections were deparaffinized and epitope retrieved by heating the tissue sections in citrate buffer (Vector Laboratories, Burlingame, CA, USA) in a microwave. After blocking with a serum-free protein blocker (Vector Laboratories) for 30 min, the tissue sections were incubated with anti-TGF-β1 primary antibodies (Appendix A). The negative control slide consisted of mouse Ig fractions (Agilent Dako, Santa Clara, CA, USA) used at the same isotype and concentration as TGF-β antibodies to determine the background intensity and staining. The tissue sections were then treated with the kit’s probe and polymer and finally developed using the BCIP/NBT (Agilent Dako) chromogen system. The slides were then mounted with Vecta Mount AQ (Vector Laboratories). An average of 19–20 view fields (40× magnification, view field area at 40× magnification was 0.116 mm^2^) were used in each of the slides to quantify TGF-β^+^ cells manually using SPOT3 live imaging software. The tissue sites for this evaluation were selected randomly and counted by two different individuals to avoid bias.

### 2.5. Flow Cytometry Analysis

LPLs were subjected to flow cytometry following the protocol reported earlier [28,32] with modification. Cells were first stained with live/dead stain (Invitrogen, Carlsbad, CA, USA). For cell surface staining, 1–1.5 × 10^6^ cells/100 µL of isolated LPLs were stained with directly conjugated monoclonal antibodies (MAbs). MAbs used for this study were anti-CD3, anti-CD4, anti-CD8, anti-CD14, anti-CD20, anti-CD45, anti-CD68, anti-CD163, anti-NKG2A, anti-IL-10, and anti-TGF-β antibodies (Appendix A). For detecting IL-10^+^ and TGF-β^+^ cells, the intracellular staining protocol was performed using BD Cytofix Cytoperm and permeabilization buffer as reported earlier [28,32]. After staining, cells were washed and fixed in BD stabilizing and fixative buffer. The flow cytometric acquisition was performed on a Becton Dickinson LSRII and Fortessa instruments. At least 50,000 events were collected from each sample by lymphocyte gating, and the data were analyzed with FlowJo software (version 10.7.1., TreeStar, Ashland, OR, USA). Singlets and live cell populations were considered for downstream analysis of all the markers.

### 2.6. Immunofluorescence Assay

Tissue sections for immunofluorescence staining with one or a combination of primary antibodies were processed as described previously [26,28,31]. In brief, deparaffinized tissue sections were processed for antigen retrieval and were successively stained with one or a combination of primary antibodies for anti-CD3, anti-CD11c, anti-CD68, anti-CD79a, anti-cytokeratin, anti-HAM56, anti-MAC387, anti-phosphorylated SMAD2/SMAD3, anti-SMAD7, and anti-TGF-βRII antibodies (Appendix A). Alexa Fluor 488 or Alexa Fluor 568 conjugated secondary antibodies (1:1000 dilution, Invitrogen) were used for staining. Anti-nuclear ToPro-3 antibodies (1 µM, Life Technologies, Carlsbad, CA, USA) or DAPI (1:5000, EMD Millipore, Burlington, MA, USA) were used for nuclear counterstaining. Labeled tissue sections were mounted using Prolong Gold antifade medium (Invitrogen). Images were captured using a TCS SP2 confocal laser scanning microscope (Leica, Wetzlar, Germany) equipped with an argon-krypton laser at 488 nm (green), a krypton laser at 568 nm (red), and a helium-neon laser at 633 nm (blue). For SMAD7 analysis, images were captured using a Nikon Ti2-E fluorescence microscopy (Nikon, Melville, NY, USA). Negative controls consisted of either omitting the primary antibody or using isotype IgG1, IgG2a, and IgG (H + L) controls. ImageJ (version 1.52; National Institutes of Health, Bethesda, MD, USA) and Adobe Photoshop (San Jose, CA, USA) were used to assign colors to the channels collected. Quantitative fluorescence densitometry was performed using ImageJ software to quantify pSMAD2/3 expression in the jejunum. Nikon NIS Elements software was used for quantification of SMAD7 mean fluorescence intensity. An average of 15–20 view fields chosen at random (20× objective, view field area at 20× magnification was 0.232 mm^2^) were quantified. The intensity of pSMAD2/3 and SMAD7 protein expression was represented as fluorescence pixel values.

### 2.7. Quantification of SMAD3 and SMAD7 Gene Expression in Jejunum Tissues

Relative quantification of gene expression was determined by real-time RT-PCR. The expression of SMAD3 and SMAD7 genes was quantified using TaqMan Gene Expression assays as described previously [1]. Briefly, total RNA was isolated from freshly cut sections of cryopreserved jejunum previously embedded in OCT compound using the Purelink FFPE RNA isolation Kit protocol (Life Technologies). The total RNA was purified using RNA Clean and concentrator-25 Kit (Zymo Research, Irvine, CA, USA) with on-column DNase I treatment (Life Technologies) followed by RNA quantification using the Synergy H4 reader (Biotek, Winooski, VT, USA). Superscript III first-strand synthesis kit (Life Technologies) was used to synthesize cDNA from total RNA. TaqMan gene expression assays Rh02621726_m1, Hs00706299_s1, and Hs00178696_m1 (Life Technologies) were employed for the quantification of TGF-β, SMAD3, and SMAD7 transcripts, respectively, using an ABI 7900HT Fast PCR System (ThermoFisher Scientific). The expression of each gene was normalized against 18S rRNA expression using TaqMan 18S rRNA Endogenous Control Assay (Life Technologies). Relative gene expression was determined among different groups using the comparative threshold cycle (C_T_) method, and fold changes in the expression were evaluated using 2^−ΔΔCT^ [33]. 

### 2.8. Statistical Analysis and Graphical Presentation

Statistical analysis and graphical presentation of the data were performed using GraphPad Prism version 9 (GraphPad Software, San Diego, CA, USA). One-way repeated ANOVA was applied to determine any statistically significant differences between the group means. Scatter dot plots are delineated as a graphical method for comparing percentages distribution of TGF-β^+^, pSMAD2/3^+^, and SMAD7^+^ cells. For the TGF-β, SMAD2/3, and SMAD7 expression analysis, the Tukey–Kramer test was applied for post hoc analysis to calculate the statistical significance between different groups. Mann–Whitney *t*-test was used for other statistical analyses. Correlation analysis between TGF-β and pSMAD2/3 expression and TGF-β and CD4 frequency in jejunum LPL were performed with a two-tailed Pearson or Spearman’s correlation methods, respectively. A *p*-value of < 0.05 was considered statistically significant.

## 3. Results 

### 3.1. Terminal Plasma Viral Loads in SIV Infected Macaques

All SIV infected RhMs had detectable plasma viral load ranging from 4.2 × 10^3^ to 1.6 × 10^8^ copies of RNA/mL of plasma. Acute SIV infected RhMs had significantly higher level of plasma viral load (ranging from 3.4 × 10^5^ to 1.6 × 10^8^ copies of RNA/mL of plasma) compared to chronically infected macaques (ranging from 4.2 × 10^3^ to 1.5 × 10^7^ copies of RNA/mL of plasma) (Table 1, *p* = 0.0002).

### 3.2. TGF-β Production Increases in the Jejunum of SIV-Infected Macaques

We quantified TGF-β^+^ cells in the jejunum of SIV-infected and uninfected macaques by immunohistochemistry staining (Figure 1). A significant increase of TGF-β^+^ cells was detected during acute (*p* = 0.0021) and chronic (*p* = 0.0002) SIV infection compared to uninfected control animals (Figure 1). However, there was no significant difference in TGF-β+ cells between acute and chronic SIV infection (*p* = 0.524). No correlation between intestinal TGF-β^+^ population and plasma viral load in SIV infected RhMs was detected.

### 3.3. Characterization of TGF-β Expressing Cell Populations in LPLs

We performed flow cytometry assay to define the intestinal cells responsible for the production of TGF-β during SIV infection. Seven animals (BC35, BD03, CL86, DE50, EJ26, FK88, and GN91) were examined for their TGF-β production in cells isolated from jejunum for up to 160 days post-SIV infection as end point analysis. Similar to immunohistochemistry, we observed an increase in TGF-β-expressing cells during SIV infection (Figure 2). Compared to preinfection and all other time points, both T (CD3^+^) and B (CD20^+^) cells showed a significant increase in TGF-β production at the 84th day post-infection (dpi) (*p* < 0.001 to 0.0001) (Figure 2B,C). We also detected increased presence of T cells (mean ± standard error, 0.71 ± 0.28 and 1.21 ± 0.44 for 0 and 14 dpi, respectively) and B cells (mean ± standard error, 0.77 ± 0.24 and 2.89 ± 1.42 for 0 and 14 dpi, respectively) positive for TGF-β 14 dpi, although this change did not reach statistical significance (Figure 2B,C). CD3^−^CD20^−^ cells produce more TGF-β from 14 dpi on (mean ± standard error, 5.47 ± 1.65) and showed a significant increase in mean TGF-β expression during both the acute phase of infection (21 dpi, mean ± standard error, 10.99 ± 2.99, *p* = 0.03) and the chronic phase of infection at 135 dpi (mean ± standard error, 14.87 ± 2.89, *p* = 0.0007) and 160 dpi (mean ± standard error, 11.48 ± 2.77, *p* = 0.02) compared to baseline values (mean ± standard error, 0.91± 0.26). The frequency of CD3^−^CD20^−^TGF-β^+^ cells was comparatively higher than the frequency of TGF-β^+^ cells among T and B cells at 14 d, 21 d, 42 d, 135 d, and 160 d with means ± standard errors ranging from 0.91 ± 0.26 to 14.87 ± 2.90 while for T and B cells the means ± standard errors on these days ranged from 0.37 ± 0.08 to 1.21 ± 0.45 and 0.46 ± 0.32 to 3.06 ± 0.73, respectively (Figure 2A–D).

We further characterized the subpopulation of TGF-β-expressing CD3^−^CD20^−^ cells based on phenotypic expression. The majority of TGF-β^+^CD3^−^CD20^−^ cells were CD68^+^ and displayed high forward and side scatter, suggesting that these TGF-β^+^ cells are bigger in size and/or in complexity or granularity, like macrophages, NKs, and/or dendritic cells (mean ± the standard error, 96.3 ± 1.5) (Figure 2E,F, contour plots, Appendix A). Only small proportion of TGF-β^+^ cells also expressed CD163, which is expressed by monocytes and macrophages (9.8%) [34], or CD14, which is expressed by monocytes (9.8%). These two markers were low or absent in the TGF-β^−^ cell population. Furthermore, the TGF-β^+^ cells were negative for the marker NKG2A expressed by cytolytic NKs [35] (Figure 2F) and only a small fraction of TGF-β^+^ cells were double positive for the immunoregulatory cytokine IL-10 (3.2%, Figure 2F). The phenotypic frequency of TGF-β^+^ cells for different cell populations ranged between 3.7 ± 0.9 and 9.7± 2.5 (mean ± the standard error), where NKG2A^+^ cells were nearly negative (mean ± standard error, 0.5 ± 0.04) for TGF-β expression (Appendix A).

A significant loss of jejunum CD4^+^ T-cells detected in all the seven RhMs starting from 14 dpi (mean percentage of CD4^+^ T cells ± the standard errors, 53.0 ± 3.0 and 10.8 ± 2.3 for day 0 and 14 dpi, respectively, *p* < 0.0001, Figure 3A,B) when compared to preinfection time point and the CD4^+^ T cell percentage remained low throughout the period of this study (Figure 3). Parametric Spearman correlation coefficient between percentages in TGF-β^+^ expression from CD3^+^ or CD20^+^ lymphocytes and percentages of CD4^+^ T cells in jejunum indicated no significant correlation between TGF-β^+^ CD3^+^/CD20^+^ lymphocytes and frequency of CD4^+^ cells was detected (*r* value from −0.113 to −0.206, *p* > 0.05, Figure 3C,D). In contrast, a highly significant negative correlation was detected between the frequency of mucosal CD3^−^CD20^−^TGF-β^+^ cells and CD4^+^ T cells in SIV infected animals (*r* = −0.582, *p* < 0.0001, Figure 3E). 

### 3.4. TGF-β Receptor II Expression Remains Unaffected during SIV Infection

In uninfected animals, TGF-βRII expression was most commonly identified in epithelial cells (Cytokeratin^+^, Figure 4A), B or plasma cells (CD79a^+^, Figure 4B), dendritic cells (CD11c^+^, Figure 4C), and macrophages (HAM56^+^, Figure 4D) [28]. Few T cells (CD3^+^, Figure 4E), and monocytes/macrophages (CD68^+^, Figure 4F) were also shown positive for TGF-βRII expression. Of note, unlike epithelial cells, the hematopoietic cells expressing TGF-βRII were not homogenously distributed but appeared to be found in small clusters. We performed immunofluorescence staining in SIV-infected jejunum tissues to measure the distribution of TGF-βRII expression in epithelial cells; however, we were unable to detect any upregulation of TGF-βRII expression in acute and chronic jejunum tissues (Appendix A). To evaluate TGF-βRII expression during SIV infection, we analyzed both frequency and median fluorescence intensity values of TGF-βRII expression in uninfected controls, SIV acutely (21 dpi) and SIV chronically (250–422 dpi) infected RhM jejunum LPLs (Figure 4G–J). We did not observe any significant differences in frequency (Figure 3I) or fluorescence intensity (Figure 4J) of TGF-βRII among the three groups of animals for CD3^+^, CD20^+^, and CD3^−^CD20^−^ cell population. 

### 3.5. An Increased Level of pSMAD2-3 Complex Suggests Persistent Activation of Signaling 

pSMAD2/3 positive cells were predominantly observed in enterocytes and lamina propria mononuclear cells (Figure 5A–C). A pronounced increase in pSMAD2/3 expression was detected in SIV-infected macaques compared to uninfected control animals (Figure 5A–C). Mean fluorescence pixel values of pSMAD2/3 in acute (mean ± standard error, 23.65 ± 1.07) and chronically (mean ± standard error, 42.09 ± 2.32) SIV-infected RhMs were significantly higher than the uninfected control animals (mean ± standard error, 8.17 ± 0.56) (Figure 5A–D) whereby the difference between acute and chronically infected animals was statistically significant (*p* < 0.0001). We also detected nuclear translocation of pSMAD2/3 protein in SIV-infected RhMs as depicted by increased light blue stains in the lamina propria cells (Figure 5). Two-tailed Pearson correlation coefficient between TGF-β^+^ cells/mm^2^ of tissues and the pSMAD2/3 immunofluorescence pixel values from control and SIV-infected RhMs indicated a significant positive correlation between the changes observed in TGF-β^+^ cells and pSMAD2/3 intensity in jejunum (*p* = 0.026, Figure 5E).

### 3.6. Upregulation of SMAD3, A Positive TGF-β/SMAD Signaling Regulator Is Associated with Disease Progression

To confirm the upregulation of pSMAD2/3 in tissues, the differential expression of a SMAD3 gene, a positive regulator of the SMAD dependent pathway, was examined in uninfected, acute, and chronically SIV-infected jejunum tissue (*n* = 4 in each group) in duplicates by TaqMan qPCR. A statistically significant increase in SMAD3 expression was detected during acute and chronic SIV infection (*p* = 0.0002) when compared to uninfected control jejunum tissues (Figure 5F). SMAD3 expression was also increased 1.5-fold more during chronic SIV infection compared to acute infection, but the upregulation was not statistically significant when compared to acute infection.

### 3.7. Downregulation of SMAD7 Expression Denotes the Failure of Negative Feedback Mechanism

We examined the expression of SAMD7 protein in the jejunum from SIV infected and uninfected healthy RhMs. We observed a significant decreased in mean fluorescence intensity of SMAD7 expression in acute (mean ± the standard errors, 405 ± 19) and chronically (mean ± the standard errors, 353 ± 21) SIV infected RhMs compared to controls (mean ± the standard errors, 620 ± 30; *p* < 0.0001, Figure 6A–D). No statistically significant difference in the mean fluorescence intensity value detected between acute and chronically SIV infected RhMs jejunum tissue (*p* = 0.27, Figure 6D). Majority of the SAMD7 protein expression was detected in the nucleus (magenta color by colocalization in Figure 6A–C). We also quantified SMAD7 mRNA expression in jejunum tissue from uninfected control and both acute and chronic SIV-infected RhMs and detected a significant downregulation of SMAD7 expression in acute and chronically SIV-infected macaques (*p* = 0.0002, Figure 6E). SMAD7 expression was also decreased 2.4-fold more during chronic SIV infection compared to acute infection, but the downregulation was not statistically significant when compared to acute infection. To determine whether there are any differences in TGF-β, SMAD3, and SMAD7 expression between male and female RhMs, we performed real-time RT-PCR on jejunum tissues obtained from 3 male (JG16, JM76, and EV39) and 3 female (AG71, DJ78, and GN70) (Table 1, Appendix A). As expected, we did not observe any significant fold changes between male and female RhMs where the mean fold change values of females relative to males ranged from 0.99, 0.74, and 1.04 for TGF-β, SMAD3, and SMAD7 genes, respectively (Appendix A).

## 4. Discussion 

Earlier studies demonstrated an increased plasma TGF-β concentration [16,36] and duodenal TGF-β gene expression from HIV-infected patients [37], which was negatively correlated with peripheral CD4 counts. In agreement with these data, in an earlier study, the level of plasma TGF-β concentration was much higher in RhMs infected with pathogenic SIV_MAC_251 compared to nonpathogenic African Green Monkey and TGF-β was thought to be a key player in regulating early immune activation and T cell responses [38]. However, in these studies, the prevalence and degree of TGF-β-producing cells and the degree of TGF-β production in intestinal tissue has not been documented. Our study suggests that the initial increase in TGF-β^+^ cells in the acute phase of infection remained elevated throughout the duration of the infection, whereby the observed increase could be due to increased production of TGF-β in resident cells or an influx of TGF-β-producing cells. We also did not detect any significant differences in TGF-β, SMAD3, and SMAD7 gene expression between male and female SIV-uninfected RhMs suggesting that their expressions in jejunum tissue are not depend on the gender of the animal. The majority of TGF-β^+^CD3^−^CD20^−^ cells were CD68^+^, which is expressed not only by monocytes and macrophages but also by non-myeloid cells like fibroblasts, endothelial cells, and neutrophil granulocytes [39,40,41]. Our data suggest that CD3^−^CD20^−^CD68^+^ cells are the key cell population responsible for mucosal TGF-β production during SIV infection. Other cells, like T and B cells, also play a role in increased mucosal TGF-β abundance; however, their role is temporary and minimal compared to the mucosal CD3^−^CD20^−^CD68^+^ cells during SIV infection. We also observed a negative correlation between frequency of mucosal CD4^+^ T cells and CD3^−^CD20^−^TGF-β^+^ cells suggesting that loss of mucosal CD4 population allowed increased representations of CD3^−^CD20^−^TGF-β^+^ cells.

After establishing that mucosal TGF-β producing cells are increased in SIV infection and are comprised of mainly non-lymphocytic/non-monocytic CD68^+^CD3^−^CD20^−^ cells, we wished to test whether this is accompanied by increased TGF-βR expression accounting for the TGF-β-associated pathology in SIV. As TGF-β exerts its cellular functions on different cells through binding to its receptors, any alteration in receptor expression may lead to the deregulation of the pathway and variation in tissue responses to the cytokine. TGF-β has been shown to mediate its pleiotropic activity in cells by interacting with TGF-βRI and TGF-βRII expressed on the cell surface [42]. We assessed the expression of TGF-βRII in jejunum from normal RhMs both by multilabel confocal microscopy and flow cytometry analysis. We focused on the expression of TGF-βRII because only TGF-βRII has high affinity for TGF-β and TGF-βRI supports downstream signaling only after the generation of heterotetrameric complex between TGF-β and TGF-βRII [43]. While TGF-βRI expression may be affected by SIV infection, the TGF-β-mediated biological effect is solely dependent on the level of TGF-βRII [44]. Our data indicated that increased TGF-β production during SIV infection was not accompanied by increased TGF-βRII expression at the mucosal surface, unlike an earlier study, which showed that TGF-β itself induced a translocation of intracellular TGF-β receptors to the cell surface and amplified its response [45]. However, that study was done in vitro with human cell lines and cannot be directly compared to our in vivo study. The loss or gain of TGF-β responses by cells that exhibit mutation or lack of TGF-βRII [46] or increased expression of TGF-β receptors [47], respectively, were reported in several previous in vitro experiments. 

In the absence of changes in TGF-βRII, we then explored whether TGF-β signaling might be altered in SIV infection. The activated TGF-βRI and RII complex helps to phosphorylate SMAD2 and SMAD3, which further bind with SMAD4 and enter the nucleus where this complex can interact with various transcription factors, coactivators, or corepressors to mediate multiple gene expression [48]. We examined the level of pSMAD2/3 protein expression and its distribution in jejunum from both SIV-infected and uninfected RhMs. Our novel findings suggest that TGF-β-mediated pSMAD2/3 upregulation in mucosal tissues provides persistent TGF-β/SMAD signaling and thus mediates TGF-β-dependent SIV pathogenesis. Increased pSMAD2/3 protein in relation to HIV infection has been previously described in an in vitro model with human pulmonary arterial smooth muscle cells in response to HIV tat protein, but only with simultaneous addition of cocaine [49]. SMAD3 has been shown to be an important mediator of many pathological fibrotic conditions [50]. Although not reported in HIV/SIV infection earlier, the upregulation of SMAD3 in mucosal tissue might play a pivotal role in HIV/SIV pathogenesis and influence the mechanisms that promote tissue fibrosis [51]. 

Unlike SMAD2 and SMAD3, SMAD7 acts as a negative regulator and maintains TGF-β signaling in balance. TGF-β itself promotes SMAD7 expression, which regulates TGF-β-dependent cellular and molecular processes in normal physiological conditions in the form of a negative feedback loop [52]. Thus, considering the increased availability of pSMAD2/3 and increased transcription of SMAD3, we expected that the inhibitory SMAD7 would be reciprocally increased to counteract the production of SMAD2/3 in SIV infection. However, our data suggest that the increased presence of TGF-β^+^ cells in the mucosa of SIV-infected RhMs also contributes to increased downstream SMAD-dependent TGF-β signaling pathway via SMAD2/3 upregulation and SMAD7 downregulation. Possibly, a lack of SMAD7 upregulation failed to prevent TGF-β-driven SMAD2/3 phosphorylation leading to an enhanced expression of the TGF-β regulated genes. This mechanism could also apply to our findings from a previous study, in which we observed increased production of mucosal IFNγ and TNFα during acute and chronic SIV infection [28]. Our data suggests that enhanced expression of inflammatory cytokines may be mediated by the lack of SMAD7 upregulation and a failure to maintain an appropriate balance between inflammatory and anti-inflammatory cytokine responses, which is crucial for the maintenance of successful immune responses in HIV infection [1]. Efficient expression of SMAD7 is strongly induced by TGF-β and several transcription factors including SMAD, Sp1, and Ap-1 under normal conditions [53,54]. HIV/SIV may also dysregulate TGF-β signaling pathway by directly or indirectly interacting with the SMAD7 molecule. In several cell types and cancer cell lines, SMAD7 regulation is induced by JAK/STAT pathway. In our earlier studies we observed significant downregulation of pSTAT3 protein expression in SIV infected jejunum tissues [28]. We also believe that the downregulation of STAT3 expression desensitize TGF-β signaling by downregulating SMAD7 expression, which is in agreement with an earlier study where SMAD7 mRNA and protein expression was significantly reduced when epidermal growth factor activity was blocked using the STAT3 knockdown cell line model [55]. There might be another potential mechanism that might regulate SMAD7 expression in HIV/SIV infection, which needs further investigation. Moreover, the effect of SIV infection on TGF-β-mediated non-SMAD signaling pathway remains to be elucidated. A schematic diagram presenting the function of TGF-β mediated effect in the intestine during SIV infection is shown in Figure 7.

Taken together, this study identified a SIV-induced increase TGF-β production contributed by the CD3^−^CD20^−^CD68^+^ cell population distinct from monocytes or macrophages and enhanced SMAD2/3 signaling in the absence of TGF-βRII changes but accompanied by a decrease of the inhibitory SMAD7 expression. This suggests that SIV infection impacts the SMAD-dependent signaling pathway of TGF-β and provides a potential framework for further study to understand the role of viral factor(s) in modulating TGF-β production and downregulating SMAD7 expression in SIV pathogenesis. These results indicate that enhanced TGF-β production by intestinal CD3^−^CD20^−^CD68^+^ cells might have negatively impacted effective mucosal immune function. Further studies are needed to determine whether regulation of TGF-β production in intestinal environment by administration of TGF-β blockers may regulate mucosal TGF-β production and help to enhance anti-SIV/HIV immune responses. 

## Figures and Tables

**Figure 1 cells-10-00806-f001:**
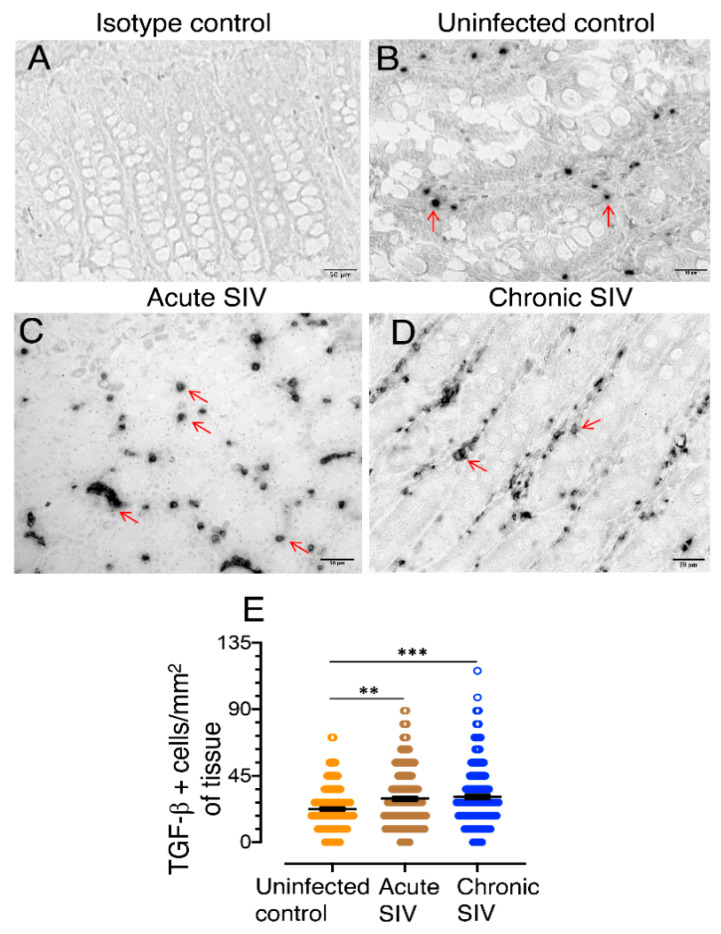
TGF-β production increases in the jejunum of SIV infected macaques detected by immunohistochemistry staining. (**A**) Representative isotype control for TGF-β showing the absence of nonspecific background staining. Representative images of TGF-β^+^ cells in SIV-uninfected (**B**), FK25, SIV acute (**C**), CF65 at 21 dpi, and SIV chronically (**D**), CL86 at 281 dpi infected macaques are shown. The red arrows are representative of TGF-β^+^ cells. (**E**) The frequency of TGF-β^+^ cells/mm^2^ of jejunum tissue with means are shown for SIV uninfected normal (*n* = 11), animals with acute SIV infection (8–21 dpi, *n* = 11), and animals with chronic SIV infection (150–422 dpi, *n* =10). An average of 19–20 fields (40× magnification) was used to quantify TGF-β^+^ cells from each animal and each value was presented by each data point. The horizontal line denotes the mean frequencies (± standard errors) of each group. Statistically significant differences between groups as analyzed with Mann–Whitney *t*-test are indicated with asterisks (**, *p* < 0.01; ***, *p* < 0.001).

**Figure 2 cells-10-00806-f002:**
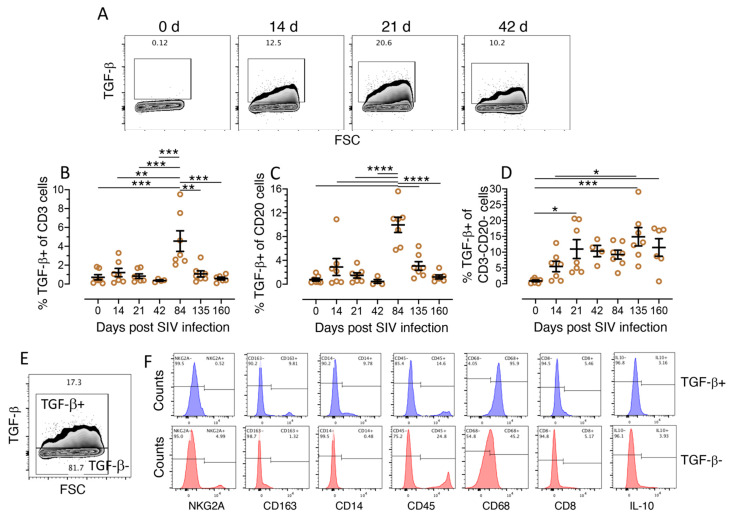
Increased production of TGF-β detected as early as 14 days after SIV infection in various types of mucosal cells. (**A**) Representative contour plots showing increased production of TGF-β in jejunum CD3^−^CD20^−^ cells detected as early as 14 d after SIV infection and remaining elevated throughout the course of the study. The percentages of TGF-β^+^ cells are shown in each box of the plot (**B**–**D**). Increased percentage of TGF-β^+^ cells (mean ± the standard error) detected in CD3^+^ T cells (**B**), CD20^+^ B cells (**C**), and CD3^−^CD20^−^ cells (**D**) isolated from the jejunum lamina propria (*n* = 7) following SIV infection. Each data point represents data from one animal for the respective time points. One-way repeated ANOVA with Tukey–Kramer’s multiple comparison test was used to calculate the significant differences among different time points. Scatter dot plots are delineated as a graphical method for comparing percentages distribution of TGF-β^+^ cells. Asterisks indicate significant differences between time points as analyzed with the Tukey–Kramer test (*, *p* < 0.05; **, *p* < 0.01; ***, *p* < 0.001; ****, *p* < 0.0001). (**E**) Contour plots for both TGF-β^+^ and TGF-β^−^ cells among CD3^−^CD20^−^ cells are shown from a SIV-infected animal 135 days post infection (BD03). TGF-β^+^ and TGF-β^−^ cells were further analyzed for the expression of different phenotypic and intracellular markers. (**F**) From this chronically SIV infected animal, jejunum CD3^−^CD20^−^TGF-β^+^ (blue histogram), and CD3^−^CD20^−^TGF-β^−^ (red histogram) cells were analyzed for expression of markers for cytotoxic NKs (NKG2A), macrophages (CD163), monocytes (CD14), pan leukocytes (CD45), monocyte/macrophages/neutrophils/fibroblasts (CD68), cytotoxic T cells (CD8), and intracellular expression of IL-10 using different monoclonal antibodies. Percentages of positive and negative population for each marker are shown at the top of each histogram.

**Figure 3 cells-10-00806-f003:**
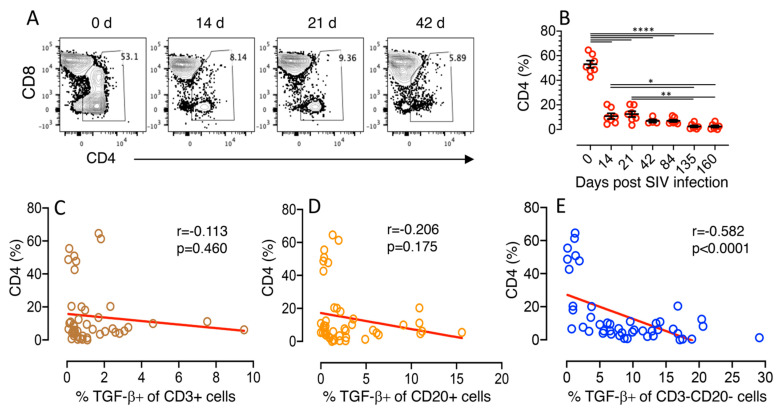
Frequency of TGF-β production by CD3^−^CD20^−^ cells negatively correlated with the frequency of mucosal CD4^+^ T cells. (**A**) Representative contour plot showing percentages of CD4^+^ T cells in jejunum lamina propria lymphocytes (LPL) where the frequency of jejunum CD4^+^ T cells decreased significantly as early as 14 days after SIV infection. The percentages of CD4^+^ T cells are shown in each gated plot. (**B**) Decreased percentage of CD4^+^ T cells (mean ± the standard errors) detected in jejunum LPL during the course of SIV infection (*n* = 7). Each data point represents data from one animal for the respective time point. Asterisks indicate significant differences between time points as analyzed with the Tukey–Kramer test (*, *p* < 0.05; **, *p* < 0.01; ****, *p* < 0.0001). Spearman’s rank correlation coefficient of determination between CD4 (%) and % TGF-β^+^ of CD3^+^ cells (**C**), CD4 (%) and % TGF-β^+^ of CD20^+^ cells (**D**), and CD4 (%) and % TGF-β^+^ of CD3^−^CD20^−^ cells (**E**) is shown for all timepoints from 7 SIV infected animals. (**E**) A significant negative correlation was detected with the expression of TGF-β from CD3^−^CD20^−^ cells and the reduction of CD4 cells in jejunum LPL.

**Figure 4 cells-10-00806-f004:**
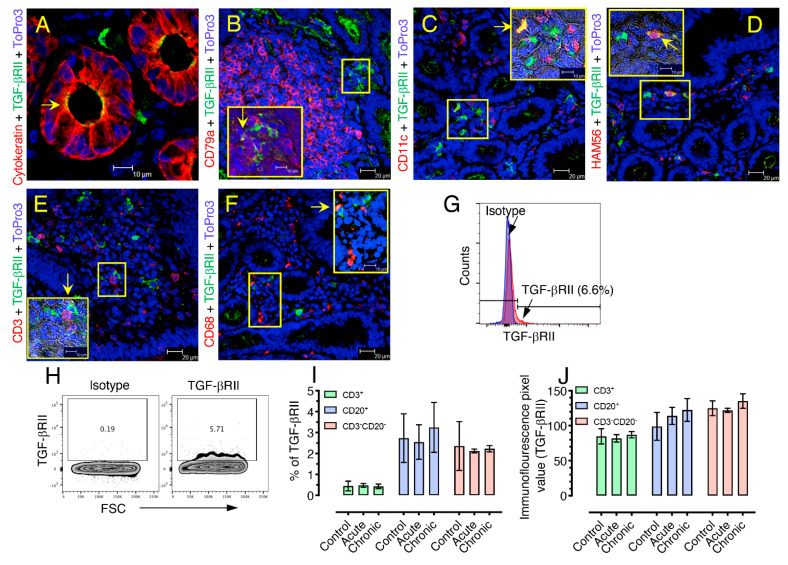
TGF-β Receptor II expression remains unaffected during SIV infection. Representative images of TGF-βRII-expressing cells in SIV-uninfected control (EV39) jejunum tissue acquired with multilabel confocal microscopy probing for epithelial cells (cytokeratin^+^, **A**), CD79a^+^ B-/plasma cells (**B**), CD11c^+^ dendritic cells (**C**), HAM56^+^ macrophages (**D**), CD3^+^ T-cells (**E**), and monocytes/macrophages (CD68^+^, **F**) are shown. Inserts in each panel show the pattern of TGF-βRII expression in conjunction with other cellular markers. Yellow arrows show colocalization of TGF-βRII and other cellular markers (as depicted by yellow color). (**G**) TGF-βRII expression in isolated CD3^−^CD20^−^ cells from jejunum was compared to the isotype control by flow cytometry assay from an uninfected control (DJ78). (**H**) Contour plots of isotype control and TGF-βRII^+^ cells are shown for CD3^−^CD20^−^ jejunum cells from an uninfected control (DJ78). The frequency of positive cells is shown in each gated box. (**I**) Frequency of TGF-βRII^+^ cells in CD3^+^, CD20^+^, and CD3^−^CD20^−^ cells in jejunum tissue from uninfected, control, and acutely and chronically infected animals (*n* = 4). (**J**) Immunofluorescence pixel values of TGF-βRII expression on CD3^+^, CD20^+^, and CD3^−^CD20^−^cells in jejunum tissue from the uninfected, control, and acutely and chronically infected animals are shown (**I**, *n* = 4).

**Figure 5 cells-10-00806-f005:**
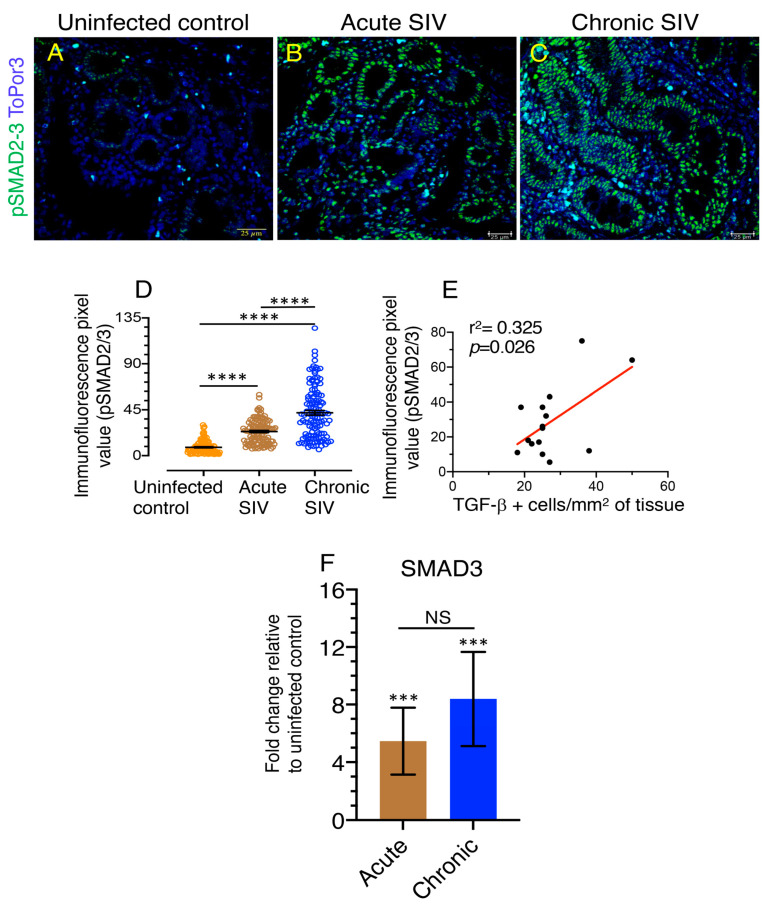
Increased levels of pSMAD2-3 complex and SMAD3 signify TGF-β/SMAD signaling dysregulation. (**A**–**C**) Representative immunofluorescent images of pSMAD2/3^+^ cells in RhM jejunum from uninfected control (**A**, GN70), animals with acute ((**B**), EM64, 21 dpi), and chronic ((**C**), CL86, 281 dpi) SIV infection. ToPro3 stains the cell nucleus. Increased pSMAD2/3^+^ cells are noticeable during acute and chronic infection compared to uninfected control RhM. (**D**) Immunofluorescence pixel values of pSMAD2/3 protein expression (indicating mean ± standard error) in jejunum from uninfected control, acute (21 dpi), and chronically (167–401 dpi) SIV-infected RhMs are shown (*n* = 6). An average of 19–20 fields (40× magnification) was used to quantify pSMAD2/3^+^ cells from each animal, and each value is presented in the graph as individual data point. (**E**) Correlation between TGF-β^+^ cells/mm^2^ of tissues and pSMAD2/3 immunofluorescence pixel values from uninfected control, acute, and chronically SIV-infected RhMs (*n* = 15). Pearson correlation coefficient analysis indicates a significant positive correlation (*p* = 0.026). (**F**) Increased fold change of SMAD3 mRNA expression were observed in jejunum from acute (21 dpi) and chronically (between 226 and 401 dpi) SIV-infected RhMs compared to uninfected control RhMs using relative RT-PCR (mean ± the standard errors, *n* = 4) but there was no statistically significant difference between acute and chronic SIV-infected RhMs. Samples were normalized against 18S rRNA expression. In all figures, asterisks indicate statistically significant differences between the respective animal groups (***, *p* < 0.001; ****, *p* < 0.0001) using Tukey–Kramer and Mann–Whitney *t*-test. NS denotes not significant.

**Figure 6 cells-10-00806-f006:**
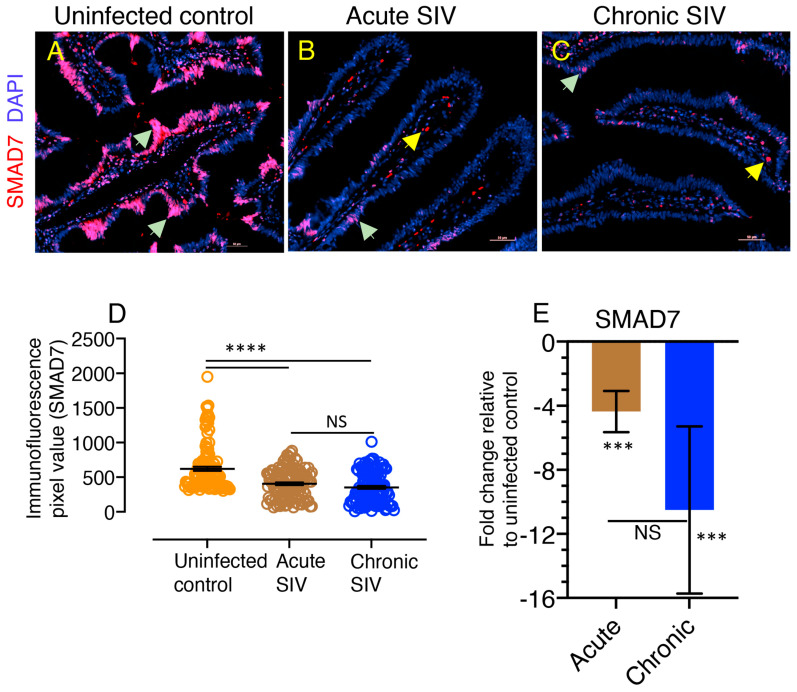
Decreased expression of SMAD7 protein and gene in SIV infection suggesting a failure of negative feedback mechanisms. (**A**–**C**) Representative immunofluorescence images of SMAD7 protein in RhM jejunum from SIV uninfected control ((**A**), EV39), animals with acute ((**B**), GI28 28 dpi), and chronic ((**C**), BD03, 167 dpi) SIV infection. DAPI stains the cell nucleus. Decreased SMAD7^+^ cells are detected during acute and chronic infection compared to uninfected control animal (magenta color by colocalization). Light green arrows show SMAD7^+^ cells where yellow arrows are autofluorescence stain (bright red color). Note that the majority of SMAD7 expression occurs in the cell nucleus (in magenta color). (**D**) Immunofluorescence pixel values of SMAD7 expression (indicating mean ± the standard errors) in jejunum from uninfected control, acute (21 dpi), and chronically (167–401 dpi) SIV infected RhMs are shown (*n* = 6). An average of 15–20 fields of 20× magnification was used to quantify SMAD7^+^ cells from each animal, and each value is presented in the graph as individual data point. (**E**) Decreased fold change of SMAD7 mRNA expression was observed in jejunum from acute (21 dpi), and chronically (between 226 and 401 dpi) SIV-infected RhMs compared to uninfected control RhMs using relative RT-PCR (mean ± the standard errors, *n* = 4). Samples were normalized against 18S rRNA expression. In all figures, asterisks indicate statistically significant differences between the respective animal groups (***, *p* < 0.001; ****, *p* < 0.0001) using Tukey–Kramer and Mann–Whitney *t*-test. NS denotes not significant.

**Figure 7 cells-10-00806-f007:**
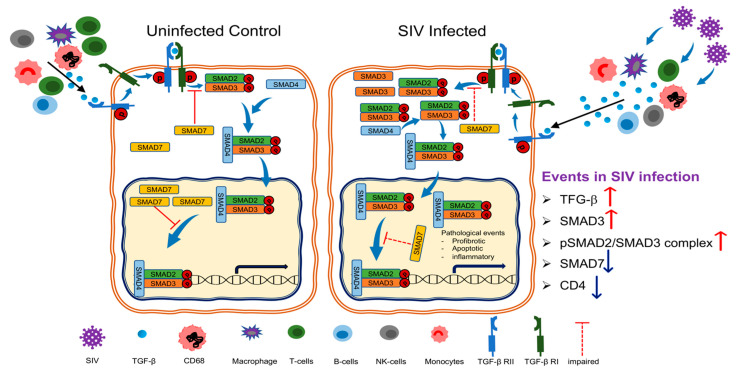
Enhanced TGF-β production detected during SIV infection. In normal, SIV uninfected control macaques active TGF-β produced from macrophages, B cells, T cells, and monocytes mediates its function after binding with the TGF-β receptor I and II (TGF-βRI and TGF-βRII), which recruits and phosphorylates TGF-βRI and SMAD proteins, a family of conserved transcription factors. Once phosphorylated, SMAD2 and SMAD3 bind with SMAD4, forming a heterodimer that enters the nucleus where it can interact with various transcription factors, coactivators, or corepressors to modulate multiple gene expressions. SMAD7 inhibits TGF-β signaling by inhibiting the formation of SMAD2/SMAD3/SMAD4 complexes or restricting binding of these complexes to DNA, thereby inhibiting transcription. During SIV infection there was a dysregulation of TGF-β production and signaling. Increased TGF-β production, along with increased SMAD3 and upregulation of pSMAD2/SMAD3 complex were detected. However, increased TGF-β production from CD3^−^CD20^−^ cells was also negatively correlated with decreased mucosal CD4 population. SMAD7 expression was also significantly downregulated in SIV infection. Collectively, the enhanced TGF-β production might have negatively impacted intestinal cytokine milieu. Regulation of mucosal TGF-β expression will help to generate effective antiviral mucosal immune responses.

**Table 1 cells-10-00806-t001:** Cumulative list of adult Indian rhesus macaques examined.

Category	Animal Number	Age (Year)	Sex ^a^	Virus	Days of Infection	Dosage (TCID_50_)	Route ^b^	Terminal Plasma Viral Load (RNA Copies/mL)
Uninfected Control	AG71	11.1	F	Nil	-	-	-	-
	DJ78	8.1	F	Nil	-	-	-	-
	EV39	6.2	M	Nil	-	-	-	-
	FK25	5.8	M	Nil	-	-	-	-
	GI92	4.2	M	Nil	-	-	-	-
	GN70	10.1	F	Nil	-	-	-	-
	GN74	13.3	F	Nil	-	-	-	-
	GT20	4.6	M	Nil	-	-	-	-
	HG11	7.1	M	Nil	-	-	-	-
	HG64	3.0	M	Nil	-	-	-	-
	JB65	7.6	M	Nil	-	-	-	-
	JG16	4.3	M	Nil	-	-	-	-
	JM76	6.3	M	Nil	-	-	-	-
Acute SIV	AV91	14.1	M	SIV_MAC_251	10	500	IV	157,190,000
	BA57	14	F	SIV_MAC_251	8	500	IV	14,288,200
	BN37	2.5	M	SIV_MAC_251	21	100	IV	340,000
	CF65	12.3	F	SIV_MAC_251	21	500	IVAG	10,100,000
	EK98	8.7	F	SIV_MAC_251	21	500	IVAG	26,800,000
	EM64	8.9	F	SIV_MAC_251	21	500	IVAG	3,840,000
	FT35	6.7	F	SIV_MAC_251	21	500	IVAG	3,540,000
	GI28	5.9	F	SIV_MAC_251	21	500	IVAG	5,830,000
	HI53	6.6	F	SIV_MAC_251	8	100	IV	3,555,700
	HN29	12.6	F	SIV_MAC_251	10	100	IV	110,000,000
	M992	16	F	SIV_MAC_251	13	500	IV	34,949,800
Chronic SIV	BC35	11.9	F	SIV_MAC_251	422	300	IVAG	428,298
	BD03	12.8	F	SIV_MAC_251	167	500	IVAG	14,788,890
	CL86	11.1	F	SIV_MAC_251	281	500	IVAG	972,889
	DE50	9.7	F	SIV_MAC_251	150	500	IVAG	304,000
	DR59	6.3	F	SIV_MAC_251	250	1000	IVAG	288,441
	EB09	6.3	F	SIV_MAC_251	250	100	IVAG	750,720
	EJ26	6.2	F	SIV_MAC_251	309	100	IV	397,806
	FK88	6.5	F	SIV_MAC_251	226	500	IVAG	2,314,583
	GN91	4.9	F	SIV_MAC_251	401	500	IVAG	4190
	HG58	9.1	F	SIV_MAC_251	288	300	IVAG	7074

^a^ F and M denote female and male respectively. ^b^ IV and IVAG denote intravenous and intravaginal route respectively. TCID_50_ represents tissue culture infectivity dose at 50%.

## Data Availability

All relevant data are included within the manuscript. The raw data are available on request from the corresponding author.

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
