# Peer review of "Enhanced Intestinal TGF-β/SMAD-Dependent Signaling in Simian Immunodeficiency Virus Infected Rhesus Macaques"

_cells, 2021, doi:10.3390/cells10040806_

Round 1

Reviewer 1 Report

Brief summary: The study presented in the manuscript by Boby et al. aimed to investigate the mechanisms by which increased production of transforming growth factor-b1 (TFG-b1) during SIV infection promotes immunosuppression. The authors identify intestinal CD3-CD20-CD68+ cells from acute and chronically SIV-infected rhesus macaques to be the primary cellular source of the increased TFG-b1 production. They further demonstrate no significant change in expression of TFG-b receptor II, but increased levels of phosphorylated SMAD2/3 proteins and SMAD3 gene expression, and decreased levels of SMAD7 gene expression in jejunum tissues. Overall, these results indicate that TFG-b1 most likely acts via regulation of SMAD2/3 signaling.

Broad comment: This study presents overall elegantly designed experiments and is nicely written. The rigor of research presented is reflected by using several approaches, such as immunofluorescence and gene expression quantification to demonstrate dysregulation of SMAD2/3 signaling. The strengths include detailed phenotypic profiling of cells that represent the source of TFG-b1, and considering multiple mechanisms while investigating how TFG-b1 may act. The data presented are consistent with the idea that TFG- b1 most likely acts via regulation of SMAD2/3 signaling. However, whether TFG-b blockers would attenuate the observed effects on SMAD2/3, or levels of production of TFG- b by the CD3-CD20-CD68+ cells, has not been investigated. Thus, the conclusion as presented in the abstract is too strong and not supported by the data. The authors phrase the same idea better in the end of the manuscript. The other major concern is unbalanced control/infection groups, which are composed of mostly male/female macaques, respectively. Unless there is evidence that there are no differences in TFG-b1 production and SMAD2/3 signaling between sexes, the conclusions of this study may be inaccurate. Some experiments have only a subset of all macaques from the study (n=4), please indicate whether these studies were balanced by sex (e.g. 4 female macaques available in the control group were used, rather than males). Gene expression quantification by RT-qPCR might be the quickest way to check whether expression of TFG-b1 and SMAD3/SMAD7 differs among males and females (especially if any samples remain for the existing control group).

Specific comments:

Methods:

1). Statistical analysis: It is not clear which analysis was used for data in Figure 2B. This is a repeated measures design of proportions/percentages (can’t assume normality of the distribution). No such methods are indicated in section 2.8 Statistical analysis and graphical presentation. Please clarify.

2). Statistical analysis: It is not clear why Tukey-Kramer test was selected for post-hoc analysis. All groups have equal number of samples (n=7), while Tukey-Kramer test is used for unequal groups.

Results:

1). Figure 1: please clarify the number of uninfected normal animals used in Figure 1D – methods and Table 1 show 13 animals in this group, this figure indicates 11. If 11, what were the reasons for exclusion?

2). In experiments that used a subset of animals (n=7 or n=4), which animals were selected or under what criteria?

3). Section 3.3: Please clarify the meaning of this sentence: “…thus, the examination day represented one animal”. Description of this experiment in methods (biopsies) and Figure 2B are consistent with the idea that multiple samples were collected from the same animal and that this is a paired design for a group of 7 animals, from whom repeated biopsies were taken.

4). Please clarify this sentence: “Compared to pre-infection, both T (CD3+) and B (CD20+) cells showed a significant increase in TGF-β production at 84th day post-infection (dpi) (p < 0.001 to 0.0001) (Figures 2B-C).” What is the actual comparison? Figure 2B has comparisons of 84 day time point to all other time points, not only pre-infection, and a range of p-values is indicated.

5). Please verify this statement: “The frequency of CD3-CD20-TGF-β+ cells was comparatively higher than the TGF-β detected from T and B cells except at 84 dpi time point (mean ± standard error, 4.55± 1.09 and 9.96± 1.27 for T and B cells, respectively Figures 2A-D).” According to the figure, the value for CD3-CD20- cells is around 10, which is comparable to B cells, but still higher than T cells, like for all other time points.

6). “The majority of the TGF-β+ cells from CD3-CD20- cells were CD68 positive” is a repetitive statement on page 8.

7). Section titles for 3.5 and 3.6 are misleading. Section 3.5 shows increased levels of phosphorylated form of SMAD2/3, but does not show persistent activation of signaling. Section 3.6 is just a RT-qPCR validation, and it does not show any relationship with disease progression. If the intent were to emphasize that there was a bigger effect in chronic compared to acute infection, then statistical analysis should have been performed to compare fold changes in acute vs chronic groups, not only acute and chronic to uninfected.

Discussion:

1). Please clarify the meaning of “magnitude” in the sentence “However, in these studies, the magnitude of TGF-β-producing cells in intestinal tissue has not been documented.”

2). “Possibly, lack of SMAD7 upregulation failed to prevent TGF-β-driven SMAD2/3 phosphorylation leading to an enhanced expression of the TGF-β regulated genes.” It would be really nice if the authors could demonstrate the effect on at least a couple of such genes (by RT-qPCR), if the RNA samples are still available.

Supplementary Figures:

1). Please check the legend of Figure S1: “Cell subsets were further gated as TGF-β+ and TGF-β+ cells based on the production of TGF-β as shown in the box.” Was this supposed to be “Cell subsets were further gated as TGF-β+ and TGF-β- cells based on the production of TGF-β as shown in the box.”?

Author Response

We greatly appreciate the critical review and thank our reviewers for their valuable suggestions. We have carefully addressed each of the comments, added new data, and revised the manuscript accordingly. Substantial changes are highlighted in yellow in the revised manuscript. Our responses to each specific comment are shown below. We hope that we have resolved all of the reviewer’s concerns.

Broad comment: This study presents overall elegantly designed experiments and is nicely written. The rigor of research presented is reflected by using several approaches, such as immunofluorescence and gene expression quantification to demonstrate dysregulation of SMAD2/3 signaling. The strengths include detailed phenotypic profiling of cells that represent the source of TFG-b1, and considering multiple mechanisms while investigating how TFG-b1 may act. The data presented are consistent with the idea that TFG- b1 most likely acts via regulation of SMAD2/3 signaling. However, whether TFG-b blockers would attenuate the observed effects on SMAD2/3, or levels of production of TFG- b by the CD3-CD20-CD68+ cells, has not been investigated. Thus, the conclusion as presented in the abstract is too strong and not supported by the data. The authors phrase the same idea better in the end of the manuscript. The other major concern is unbalanced control/infection groups, which are composed of mostly male/female macaques, respectively. Unless there is evidence that there are no differences in TFG-b1 production and SMAD2/3 signaling between sexes, the conclusions of this study may be inaccurate. Some experiments have only a subset of all macaques from the study (n=4), please indicate whether these studies were balanced by sex (e.g. 4 female macaques available in the control group were used, rather than males). Gene expression quantification by RT-qPCR might be the quickest way to check whether expression of TFG-b1 and SMAD3/SMAD7 differs among males and females (especially if any samples remain for the existing control group).

We like to thank the reviewer for the overall appreciation of our manuscript. We agree with the reviewer’s comments regarding the conclusion statement in the abstract and discussion section. We have now revised the text.

Thank you for your concern regarding the baseline expression of TGF-b, SMAD3, and SMAD7 in our experimental groups. We were able to perform real-time RT-PCR for those genes from male and female RhMs (n=3 per group) and did not find any significant differences in gene expression between male and female RhMs for those genes (see new Supplemental Figure S4). Therefore, we can conclude that their expressions are not sex dependent.

Specific comments:

Methods:

1). Statistical analysis: It is not clear which analysis was used for data in Figure 2B. This is a repeated measures design of proportions/percentages (can’t assume normality of the distribution). No such methods are indicated in section 2.8 Statistical analysis and graphical presentation. Please clarify.

Thanks for this comment. We have now clarified the method in section 2.8 as well as in the figure 2 legend.

2). Statistical analysis: It is not clear why Tukey-Kramer test was selected for post-hoc analysis. All groups have equal number of samples (n=7), while Tukey-Kramer test is used for unequal groups.

We agree with the reviewer’s statement on Tukey-Kramer. However, at 42 days post infection time point the number of samples were 4. Therefore, we have used Tukey-Kramer test for this analysis.

Results:

1). Figure 1: please clarify the number of uninfected normal animals used in Figure 1D – methods and Table 1 show 13 animals in this group, this figure indicates 11. If 11, what were the reasons for exclusion?

Table 1 is the cumulative list of all animals. We have only used up to 11 animals per group for uninfected and acute SIV animals and 10 from chronic SIV animals for the experiment depicted in the Figure 1 due to the unavailability of all animals at the beginning of the study (in Figure 1).

2). In experiments that used a subset of animals (n=7 or n=4), which animals were selected or under what criteria?

Thank you for pointing out this unclarity. We have now included the animals that have been used for that experiment. The data in Figure 2 was generated from 7 animals in a longitudinal study using several timepoints. At 42 days post infection data were collected from only 4 out of 7 animals.

Four animals per group were used for SMAD3 and SMAD7 real-time RT-PCR analysis and we have randomly selected those animals. We have now included the animal information in the revised text.

3). Section 3.3: Please clarify the meaning of this sentence: “…thus, the examination day represented one animal”. Description of this experiment in methods (biopsies) and Figure 2B are consistent with the idea that multiple samples were collected from the same animal and that this is a paired design for a group of 7 animals, from whom repeated biopsies were taken.

We apologize for this confusion and have revised the sentence.

4). Please clarify this sentence: “Compared to pre-infection, both T (CD3+) and B (CD20+) cells showed a significant increase in TGF-β production at 84th day post-infection (dpi) (p < 0.001 to 0.0001) (Figures 2B-C).” What is the actual comparison? Figure 2B has comparisons of 84 day time point to all other time points, not only pre-infection, and a range of p-values is indicated.

We apologize for the unclarity and have corrected the sentence by including “and all other time points”.

5). Please verify this statement: “The frequency of CD3-CD20-TGF-β+ cells was comparatively higher than the TGF-β detected from T and B cells except at 84 dpi time point (mean ± standard error, 4.55± 1.09 and 9.96± 1.27 for T and B cells, respectively Figures 2A-D).” According to the figure, the value for CD3-CD20- cells is around 10, which is comparable to B cells, but still higher than T cells, like for all other time points.

We have corrected the sentence.

6). “The majority of the TGF-β+ cells from CD3-CD20- cells were CD68 positive” is a repetitive statement on page 8.

Thank you.. We have deleted that sentence.

7). Section titles for 3.5 and 3.6 are misleading. Section 3.5 shows increased levels of phosphorylated form of SMAD2/3, but does not show persistent activation of signaling. Section 3.6 is just a RT-qPCR validation, and it does not show any relationship with disease progression. If the intent were to emphasize that there was a bigger effect in chronic compared to acute infection, then statistical analysis should have been performed to compare fold changes in acute vs chronic groups, not only acute and chronic to uninfected.

We apologize for this oversight and thank you for the opportunity to improve our data. We have reanalyzed the data and found significant differences of pSMAD2/3 during acute infection and chronic infection compared to uninfected control macaques and between acute and chronic infection with highest values for chronic infection. The changes at the mRNA level were similar though the difference between acute and chronic infection did not reach statistical significance.   Thus, the upregulation of SMAD3 appears to be persistent. The revised analysis and figure are included in the manuscript (revised figure 5).

Discussion:

1). Please clarify the meaning of “magnitude” in the sentence “However, in these studies, the magnitude of TGF-β-producing cells in intestinal tissue has not been documented.”

We apologize for the unclarity and we  have clarified the sentence.

2). “Possibly, lack of SMAD7 upregulation failed to prevent TGF-β-driven SMAD2/3 phosphorylation leading to an enhanced expression of the TGF-β regulated genes.” It would be really nice if the authors could demonstrate the effect on at least a couple of such genes (by RT-qPCR), if the RNA samples are still available.

We thank the reviewer for this valuable comments. Unfortunately, we do not have additional RNA samples left over to perform additional study to determine those upregulated genes and will be performed in a future study.

Supplementary Figures:

1). Please check the legend of Figure S1: “Cell subsets were further gated as TGF-β+ and TGF-β+ cells based on the production of TGF-β as shown in the box.” Was this supposed to be “Cell subsets were further gated as TGF-β+ and TGF-β- cells based on the production of TGF-β as shown in the box.”?

We apologize for this typo. We have now fixed that sentence.

Reviewer 2 Report

In this study, Boby and colleagues have identified the cellular source of TGF-b and the underlying downstream events that are potentially responsible for pathological manifestation of TGF-β in HIV/SIV infection. The authors identified CD3-CD20-CD68+ cells as the main source of increased intestinal TGF- β in both acute and chronically infected rhesus macaques. The increased TGF- β production was not accompanied by concomitant increase in the expression of TGF- βRII. Downstream of TGF- βRII, there was an increase in phosphorylation of SMAD2/3, upregulation of SMAD3 expression and downregulation of SMAD7 expression. The authors conclude that increased TGF- β in SIV manifests its pathological effect via deregulation of SMAD-dependent signaling pathway and propose potential use of TGF- β blockers as a therapeutic approach to counteract mucosal TGF- β in HIV/SIV.

Major points:

  1. The authors have convincingly shown increased levels of intestinal TGF- β and deciphered the downstream signaling mechanisms. What is lacking though is a correlation between enhanced TGF- β and its biological implication(s). In this regard have the authors done the following?        
  1. Looked at the CD4+ T cell counts in both acute and chronically infected RhMs (evidence have suggested that TGF- β concentration negatively correlates with peripheral CD4+ T cell counts in HIV-1 infected patients)
  2. Looked at the health/integrity of enterocytes (literature has shown that increased SMAD3 can cause tissue fibrosis and TGF- β promotes epithelial cell survival)    
  1. Table 1 – What was the basis for administering different dosages to animals?
  2. Table 1 – How was TCID50 determined?
  3. Table 1 – Any reason for different routes of virus administration (IV vs IVAG)?
  4. Figure 4 - It would be informative to know which sub-populations of the lamina propria cells showed higher expression of pSMAD2/3. Are they the same cells that express more TGF-β?
  5. Figure 4F and 4G – Quantitation of SMAD3 and SMAD7 at protein level (similar to pSMAD2/3) would be more informative. Could you please indicate why their expression was not quantified at protein level?

Minor:

  1. Line 3 (introduction) – ‘pleiotropic’ instead of ‘pleotropic’
  2. Figure 1 legend – SIV uninfected control is panel B, SIV acute is panel C, SIV chronic is panel D and frequency of TGF-β cells/mm2 is panel E.
  3. Figure 2E and 2F text – Fig. 2E and 2F are from one chronically infected animal while the data in supplementary Fig.2 are from three acutely infected animals. Please specify this in the text.
  4. Figure 3 legend – Please correct the following in the figure legend.

        CD79a+ is panel B

        CD11c+ is panel C

        HAM56+ macrophages is panel D

        CD3+ T cells is panel E

        CD68+ cells is panel F

  1. Discussion section – Can you speculate on possible explanations for the failure of SMAD7 upregulation (negative feedback) in the presence of high TGF-β?

Author Response

We greatly appreciate the critical review and thank our reviewers for their valuable suggestions. We have carefully addressed each of the comments, added new data, and revised the manuscript accordingly. Substantial changes are highlighted in yellow in the revised manuscript. Our responses to each specific comment are shown below. We hope that we have resolved all of the reviewer’s concerns.

Major points:

  1. The authors have convincingly shown increased levels of intestinal TGF- β and deciphered the downstream signaling mechanisms. What is lacking though is a correlation between enhanced TGF- β and its biological implication(s). In this regard have the authors done the following?        
  1. Looked at the CD4+ T cell counts in both acute and chronically infected RhMs (evidence have suggested that TGF- β concentration negatively correlates with peripheral CD4+ T cell counts in HIV-1 infected patients)

We thank the reviewer for this valuable suggestion. We have now compared the mucosal CD4 and TGF-b population and presented the data as new Figure 3. We have only detected a negative correlation between CD3-CD20- TGF-b + cells and mucosal CD4+ T cells. The data and results have been included. The previous Figures 3 and 4 were renumbered as Figure 4 and 5, respectively.

  1. Looked at the health/integrity of enterocytes (literature has shown that increased SMAD3 can cause tissue fibrosis and TGF- β promotes epithelial cell survival)

This is an important question. In fact, we have shown that TGF-b-regulated pAKT and IFNg expressions were associated with epithelial cell survival in rhesus macaque colon explants and suggest a potential role of mucosal TGF-b in regulating intestinal homeostasis and epithelial cell integrity (Pahar et al., 2015). A balanced TGF-b expression is beneficial for cell survival but it’s increased expression also promotes apoptosis through the upregulation of different pro apoptotic signaling pathways including Bim (Ramesh et al., 2009). In our earlier study we have also shown that the loss of epithelial cells and tight junction proteins happen very early in SIV infection (Pan et al., 2014). In this manuscript we have not measured enterocyte apoptosis and expression of tight junction which is beyond the scope of this manuscript.

  1. Table 1 – What was the basis for administering different dosages to animals?

Different virus dosage was used in this study. In our earlier studies we have not detected any association between virus dosage, CD4 depletion and viral load in animals (Pahar et al., 2016; Pahar et al., 2019; Pan et al., 2014). We have now included the explanation in the materials and methods section.

  1. Table 1 – How was TCID50 determined?

We have included a description of how TCID50 was determined.

  1. Table 1 – Any reason for different routes of virus administration (IV vs IVAG)?

The route of infection has no impact on the SIV plasma viral load, CD4 depletion and pathogenesis (Pahar, 2016). We have also not observed any significant difference in either RNA or DNA positive cells in tissues when animals were grouped by sex (male or female), viral dose, or route of virus exposure in our study (Pahar et al., 2019). We have included an explanation in the materials and methods section. 

  1. Figure 4 - It would be informative to know which sub-populations of the lamina propria cells showed higher expression of pSMAD2/3. Are they the same cells that express more TGF-β?

pSMAD2/3 is expressed by both cells from lamina propria as well as epithelial cells (Pahar, 2015). We have seen increased expression of pSMAD2/3 detected in epithelial cells compared to lamina propria cells after SIV infection where as TGF-β cells were predominantly expressed in the cells from lamina propria. It is definitely possible that same cells are positive for pSMAD2/3 and TGF-β. However due to the unavailability of fluorochrome labeled pSMAD2/3 antibodies we can not perform flow cytometry to determine the coexpression of TGF-β and pSMAD2/3.

  1. Figure 4F and 4G – Quantitation of SMAD3 and SMAD7 at protein level (similar to pSMAD2/3) would be more informative. Could you please indicate why their expression was not quantified at protein level?

Thank you for your valuable comments. We have now included new data related to SMAD7 protein expression in the revised figure 6. Note, that we have detected SMAD7 downregulation at protein and gene level during SIV infection. 

Minor:

  1. Line 3 (introduction) – ‘pleiotropic’ instead of ‘pleotropic’

We apologize for the typo and have corrected this.

  1. Figure 1 legend – SIV uninfected control is panel B, SIV acute is panel C, SIV chronic is panel D and frequency of TGF-β cells/mm2 is panel E.

We apologize for the oversight and have corrected this.

  1. Figure 2E and 2F text – Fig. 2E and 2F are from one chronically infected animal while the data in supplementary Fig.2 are from three acutely infected animals. Please specify this in the text.
  2. We apologize for the oversight and have corrected thisFigure 3 legend – Please correct the following in the figure legend.

        CD79a+ is panel B

        CD11c+ is panel C

        HAM56+ macrophages is panel D

        CD3+ T cells is panel E

        CD68+ cells is panel F

            We apologize for this mislabelling. It has now been corrected.

  1. Discussion section – Can you speculate on possible explanations for the failure of SMAD7 upregulation (negative feedback) in the presence of high TGF-β?

Thank you for your valuable comments. We have now included a paragraph to address a possible explanation in the revised manuscript.

References:

Pahar, B., Kenway-Lynch, C.S., Marx, P., Srivastav, S.K., LaBranche, C., Montefiori, D.C., and Das, A. (2016). Breadth and magnitude of antigen-specific antibody responses in the control of plasma viremia in simian immunodeficiency virus infected macaques. Virol J 13, 200.

Pahar, B., Kuebler, D., Rasmussen, T., Wang, X., Srivastav, S.K., Das, A., and Veazey, R.S. (2019). Quantification of Viral RNA and DNA Positive Cells in Tissues From Simian Immunodeficiency Virus/Simian Human Immunodeficiency Virus Infected Controller and Progressor Rhesus Macaques. Front Microbiol 10, 2933.

Pahar, B., Pan, D., Lala, W., Kenway-Lynch, C.S., and Das, A. (2015). Transforming growth factor-beta1 regulated phosphorylated AKT and interferon gamma expressions are associated with epithelial cell survival in rhesus macaque colon explants. Clin Immunol 158, 8-18.

Pan, D., Kenway-Lynch, C.S., Lala, W., Veazey, R.S., Lackner, A.A., Das, A., and Pahar, B. (2014). Lack of interleukin-10-mediated anti-inflammatory signals and upregulated interferon gamma production are linked to increased intestinal epithelial cell apoptosis in pathogenic simian immunodeficiency virus infection. J Virol 88, 13015-13028.

Ramesh, S., Wildey, G.M., and Howe, P.H. (2009). Transforming growth factor beta (TGFbeta)-induced apoptosis: the rise & fall of Bim. Cell Cycle 8, 11-17.

Round 2

Reviewer 1 Report

The authors have been very responsive to the previous reviews and have significantly improved the manuscript. Thank you for being diligent in acquiring additional data to make sound conclusions.